# The Phylogeny and Metabolic Potentials of an Aromatics-Degrading *Marivivens* Bacterium Isolated from Intertidal Seawater in East China Sea

**DOI:** 10.3390/microorganisms12071308

**Published:** 2024-06-27

**Authors:** Chengwen Sun, Zekai Wang, Xi Yu, Hongcai Zhang, Junwei Cao, Jiasong Fang, Jiahua Wang, Li Zhang

**Affiliations:** 1Shanghai Engineering Center of Hadal Science and Technology, College of Marine Sciences, Shanghai Ocean University, Shanghai 201306, China; sunchengwen2022@163.com (C.S.); M220200648@shou.edu.cn (Z.W.); xyu@shou.edu.cn (X.Y.); zhanghongcai2021@163.com (H.Z.); jwcao@shou.edu.cn (J.C.); jsfang@shou.edu.cn (J.F.); 2Laboratory for Marine Biology and Biotechnology, Qingdao National Laboratory for Marine Science and Technology, Qingdao 266237, China

**Keywords:** marine bacteria, *Marivivens*, genomic comparison, lignocellulosic material biodegradation

## Abstract

Lignocellulosic materials, made up of cellulose, hemicellulose, and lignin, constitute some of the most prevalent types of biopolymers in marine ecosystems. The degree to which marine microorganisms participate in the breakdown of lignin and their impact on the cycling of carbon in the oceans is not well understood. Strain LCG002, a novel *Marivivens* species isolated from Lu Chao Harbor’s intertidal seawater, is distinguished by its ability to metabolize lignin and various aromatic compounds, including benzoate, 3-hydroxybenzoate, 4-hydroxybenzoate and phenylacetate. It also demonstrates a broad range of carbon source utilization, including carbohydrates, amino acids and carboxylates. Furthermore, it can oxidize inorganic gases, such as hydrogen and carbon monoxide, providing alternative energy sources in diverse marine environments. Its diversity of nitrogen metabolism is supported by nitrate/nitrite, urea, ammonium, putrescine transporters, as well as assimilatory nitrate reductase. For sulfur assimilation, it employs various pathways to utilize organic and inorganic substrates, including the SOX system and DSMP utilization. Overall, LCG002’s metabolic versatility and genetic profile contribute to its ecological significance in marine environments, particularly in the degradation of lignocellulosic material and aromatic monomers.

## 1. Introduction

Lignin is a complex, amorphous aromatic polymer and the second most abundant natural organic polymer on Earth, following cellulose [1]. It constitutes one of the three primary components of lignocelluloses, alongside cellulose and hemicelluloses, which form plant cell walls [2]. Lignin is composed mainly of three types of phenolic precursors: p-hydroxyphenyl, guaiacol, and syringyl units. These units are interconnected through carbon–carbon (C–C) and carbon-oxygen (C–O) bonds created by radical coupling reactions [2,3]. The molecular complexity and lack of a regular repeating structure make lignin resistant to degradation by conventional methods [4,5]. However, certain terrestrial specialized microorganisms, particularly fungi and bacteria, have evolved the ability to break down lignin through a process known as microbial biocatalysis [5,6]. These organisms secrete enzymes that facilitate the degradation of lignin [3,7,8,9,10], including lignin peroxidase (LiP), manganese peroxidase (MnP), laccase (Lac) and dye-decolorizing peroxidases (DyP). During this breakdown, aromatic compounds are released [11,12,13,14], including phenolic acids and simple aromatic hydrocarbons, which are then utilized by a variety of bacteria and fungi [15,16,17,18,19,20]. 

River systems convey substantial quantities of plant-derived organic material to the marine environment, contributing to a carbon flow roughly equivalent to 0.7% of the Earth’s terrestrial primary productivity [21], which could serve as essential nutrients that drive metabolic activities in marine ecosystems [7,8,16,22,23]. Nevertheless, a limited number of bacteria capable of degrading lignin have been isolated from marine environments, and the degree to which marine microorganisms participate in the breakdown of lignocellulosic substances and their impact on the marine carbon cycle is not yet fully understood.

*Marivivens* species, belonging to the family Rhodobacteraceae, are characterized by several distinct physiological features that enable them to thrive in marine environments. The genus was first described with *Marivivens donghaensis* in 2016 [24], characterized by its unique phylogenetic lineage. Since then, the genus has expanded to include *Marivivens niveibacter* [25], isolated from tropical mangrove seawater, and *Marivivens aquimaris* [26], isolated from the Yellow Sea coast of South Korea. Recently, *Aestuarium zhoushanense* and *Paradonghicola geojensis* were also reclassified as *Marivivens* species after taxonomic revisions [27]. Morphologically, *Marivivens* species are typically aerobic, Gram-negative, non-flagellated, non-motile and rod-shaped [24,25,26]. They exhibit a wide range of temperature and pH tolerance, allowing them to grow optimally at temperatures between 25 °C and 40 °C, and at pH levels ranging from 6.0 to 10.0 [26]. However, the scarcity of research has resulted in an incomplete comprehension of the genomic traits, metabolic capabilities, and ecological roles of the bacteria classified under the genus *Marivivens*.

In this study, we present a novel strain, *Marivivens* sp. LCG002, from the lignin-enriched intertidal seawater of Lu Chao Harbor in the East China Sea, along with its genome sequences, as the second complete genome of the genus *Marivivens*. By employing genomic comparison, reconstruction of metabolic pathways and biological experiments, we have comprehensively investigated the metabolic capabilities of strain LCG002, with a particular focus on its potential to degrade lignin, aromatic acids, carbohydrates, proteins and carboxylates. Our study highlights that *Marivivens* species could serve as an important player in the microbial transformation of complex organic matter in marine ecosystems. 

## 2. Materials and Methods

### 2.1. Sample Description and Bacterial Collection

Surface water specimens were gathered from the intertidal region of Lu Chao Harbor, located within the East China Sea (30.845° N, 121.848° E) in October 2022. These samples were incorporated into the culture medium at a dilution rate of 1:100, where the medium was composed of artificial seawater (ASW) supplemented with 5 mM of lignin as the sole carbon source. Following a three-week period of incubation at ambient temperature, the microbial enrichments were further diluted to a ratio of 1:1000 using ASW and then inoculated onto marine 2216E-agar plates, which included 1000 mL seawater, 5 g Peptone, 1 g Yeast Extract and 15 g Agar, pH 7.5–7.6). The inoculated plates were incubated at a temperature of 20 °C for a duration of two weeks. Subsequently, colonies were isolated and purified through the method of streaking. The particular strain designated as LCG002 (=MCCC 1K08953) was routinely cultivated on 2216E medium under aerobic conditions and, for long-term preservation, it was stored at −80 °C in a liquid medium that included a 20% (*v*/*v*) concentration of glycerol. 

For this study, the formula of the artificial seawater (ASW) consisted of 52 g/L of NaCl, 10 g/L of MgCl_2_·6H_2_O, 8 g/L of Na_2_SO_4_, 2.8 g/L of CaCl_2_·2H_2_O, 1 g/L of KCl, 0.6 g/L of NH_4_Cl, 0.2 g/L of KH_2_PO_4_ and 2 mM of NaHCO_3_. The ASW was sterilized by autoclave and later supplemented with trace elements and vitamin mixture, both at a dilution rate of 1:1000. The trace elements contained 30 mg/L of FeCl_3_·6H_2_O, 2 mg/L of MnCl_2_·4H_2_O, 0.23 mg/L of ZnSO_4_·7H_2_O, 0.2 mg/L of CoCl_2_·6H_2_O, 0.1 mg/L of Na_2_MoO_4_·2H_2_O, 0.2 mg/L of Na_2_SeO_3_ and 0.2 mg/L of NiCl_2_·6H_2_O, while the supplementary vitamin mixture included 1.8 g/L of thiamine, 1 g/L of myoinositol, 100 mg/L of pyridoxine, 98.4 mg/L of nicotinic acid, 80 mg/L of 4-aminobenzoic acid, 30 mg/L of pantothenic acid, 1 mg/L of folate, 1 mg/L of cobalamin and 0.1 mg/L of biotin. Both trace elements and vitamin mixture were filtrated through 0.1 µm filters before their addition. 

### 2.2. Microbial Utilization of Carbohydrates, Carboxylates and Peptone

The capacity to metabolize a variety of carbohydrates, carboxylates and peptone was evaluated in triplicate on ASW agar plates with single supplements, including fructose, sucrose, raffinose, nystose, cellulose, sodium formate, sodium acetate, sodium propionate or peptone, each serving as the exclusive carbon source. The final concentrations of peptone and cellulose were 0.5 g/L, and those of other carbohydrates were 5 mM [28]. Cultures of strain LCG002 in the exponential phase, initially grown on 2216E medium (OD600 = 0.2), were subjected to a washing process, which involved three rounds of centrifugation and resuspension in equal volumes of ASW to eliminate the remaining 2216E. Notably, sodium tungstate was added at the concentration of 0.1mg/L into ASW medium with sodium formate. Subsequently, 50 µL of the bacterial solution was coated onto separated plates with a distinct carbon source in triplicate and incubated at 30 °C. The growth conditions on these plates were presented via photographs after seven days. 

### 2.3. Microbial Degradation of Lignin and Aromatic Acids

The lignin decomposition capacity of the microorganism was assessed in triplicate using a 100 mL volume of Artificial Seawater (ASW) liquid medium, supplemented with lignin as the sole carbon source at a concentration of 5 mM. The cultures of strain LCG002 in the exponential phase were centrifugally washed with ASW liquid medium three times and then inoculated into lignin-supplemented and lignin-free ASW liquid medium, respectively, at the cultivation temperature of 30 °C. Live bacteria were respectively counted at day 0 and day 7 through the utilization of SYTO 9/PI live/dead bacteria double strain kit. The degradation of lignin by strain LCG002 was evaluated by comparing the colony numbers of strain LCG002 in the control and experimental groups. In addition, the potential for lignin degradation was verified by an experiment of aniline blue decolorization, which was performed via inoculating strain LCG002 on 2216E-agar plates with 3% aniline blue added and cultivating at 30 °C. The decolorized zones around the colonies on the agar medium were observed after 2 days. 

The capacity of the microbe to metabolize aromatic acids was evaluated in triplicate using 100 mL of ASW liquid medium supplemented with benzoic acid, 3-hydroxybenzoic acid or 4-hydroxythylbenzoic acid, serving as the sole carbon source. The final concentration of benzoic acid was set at 0.8 mM, while that of 3-hydroxybenzoic acid and 4-hydroxythylbenzoic was 1 mM. Exponential-phase cultures of strain LCG002 were washed via centrifugation with ASW liquid medium three times and then aliquoted into distinct ASW liquid media containing the various aromatic acids for cultivation at 30 °C. Aliquots of 50 µL from the bacterial cultures at both 0 and 7 days post-inoculation were respectively coated onto marine 2216E agar plates in triplicate and incubated at 30 °C. The utilization of aromatic acids by strain LCG002 was verified by comparing the colony numbers of strain LCG002 on the plates between initial (day 0) and substantial (day 7) time points. 

### 2.4. Genomic DNA Extraction, Sequencing and Extraction

The genomic DNA of strain LCG002 was extracted following the protocol described by Fang et al. [29]. Thereafter, the genome sequencing of strain LCG002 was conducted by MajorBio (Shanghai Majorbio Bio-pharm Technology Co., Ltd., Shanghai, China), employing a hybrid sequencing strategy that integrated the PacBio RS II Single Molecule Real-Time (SMRT) (Pacific Biosciences, Menlo Park, CA, USA) with the Illumina HiSeq 2500 platforms (Illumina Inc., San Diego, CA, USA). 

In the Illumina sequencing protocol, approximately 1 µg of genomic DNA was sheared into fragments within the range of 400–500 bp using the Covaris M220 Focused Acoustic Shearer (Covaris, Woburn, MA, USA), following the protocol provided by the manufacturer. After fragmentation, the DNA segments were employed to assemble the Illumina sequencing libraries, facilitated by the NEXTFLEX Rapid DNA-Seq kit (NEXTFLEX, San Jose, CA, USA). 

For Pacific Biosciences sequencing, a 15-µg aliquot of DNA was prepared and centrifuged in a Covaris g-TUBE at 6000 RPM for a duration of 60 s through the utilization of an Eppendorf 5424 centrifuge (Eppendorf, Hamburg, Germany). Subsequently, the DNA fragments underwent purification, end-repairing and were ligated with SMRTbell sequencing adapters in accordance with Pacific Biosciences’ protocols. The sequencing library generated was further purified for three rounds using 0.45× volume of Agencourt AMPure XP beads (Agencourt, Scottsdale, AZ, USA), following the manufacturer’s instruction. Next, an insert library in the size of ~10 kb was prepared and sequenced on one SMRT cell via the standard methods. 

Data obtained from both PacBio and Illumina sequencing platforms were leveraged for the bioinformatics analysis. The complete genome sequence was reconstructed by merging the sequencing reads derived from both PacBio and Illumina technologies. The preliminary raw image data were converted into sequence data via the base calling process, yielding the initial raw reads, which were then stored in the FASTQ file format. Subsequently, a quality trimming procedure was implemented to remove low-quality data segments, utilizing a statistical approach for quality assessment. The refined reads were assembled into contiguous sequences through the hierarchical genome assembly process (HGAP) and the canu software [30]. The final stage of the process involved manually verifying and completing the circularization of the genome, culminating in a finished genome sequence featuring a single, unbroken chromosome. Ultimately, the assembled genome from PacBio was refined using the Illumina reads with Pilon [31] to correct any potential errors. The full genomic sequence of the *Marivivens* sp. LCG002 strain has been lodged with the GenBank database, assigned the accession number CP 127165.1. 

### 2.5. Gene Annotation and Genomic Comparison

For the identification and characterization of open reading frames (ORFs), we utilized the annotation pipeline [32] provided by the NCBI prokaryotic genome. Subsequently, the deduced protein sequences were subjected to alignment against the Clusters of Orthologous Groups of proteins (COG) [33] and TransporterDB 2.0 [34] databases employing the BLASTp algorithm (version 2.9.0) with the following parameters set for significance: an identity threshold of 50%, a query coverage of 80%, an e-value cutoff of 1 × 10^−5^, and a minimum score of 40. BlastKOALA [35] was applied to facilitate the assignment of Kyoto Encyclopedia of Genes and Genomes (KEGG) functional annotations. Furthermore, IslandViewer 4 [36] was engaged to prognosticate the existence and precise positioning of genomic islands embedded within the genomic sequence. 

To classify the protein families of strain LCG002 alongside its phylogenetically affiliated strains, we deployed the local OrthoMCL version 2.0.9 [36] for clustering. The clustering procedure was conducted with specific cutoff criteria, including a sequence identity of 50%, a query coverage of 50%, an E value threshold of 1 × 10^−10^, a minimum score of 40, and an MCL inflation value set at 1.5. Protein families detected exclusively in a single strain were deemed to be strain-specific. The average nucleotide identity (ANI) and average amino acid identity (AAI) were determined using fastANI [37] and CompareM (https://github.com/dparks1134/CompareM, accessed on 10 December 2023), respectively, with their standard parameter settings, while the genome-to-genome distance (DDH) was ascertained through the application of GGDC 3.0 [38]. 

### 2.6. Phylogenetic Analysis

For elucidating the phylogenetic relationships of strain LCG002 and its associated strains, we utilized a dataset of 120 conserved bacterial marker genes as delineated by the Genome Taxonomy Database (GTDB). Initially, the genomic sequences encoding these 120 marker proteins were identified using the GTDB-Tk toolkit (reference database version: Release 07-RS207) [39]. We then proceeded to align these sequences individually using Clustal Omega [40]. Following alignment, we manually removed gaps and adjusted for any missing marker proteins in certain genomes by inserting an equivalent number of “-” to match the length of the sequences’ post-gap removal. Subsequently, the alignments, now devoid of gaps, were concatenated for each marker protein [41]. The resultant concatenated alignment was used to construct a phylogenetic tree employing the neighbor-joining algorithm as implemented in FastTree2 [42]. To assess the robustness of the tree, a bootstrap test with 1000 iterations was conducted. Finally, the phylogenetic tree was visualized using the Interactive Tree Of Life (iTOL) software [43]. 

## 3. Results and Discussions

### 3.1. Description of Strain LCG002

Intertidal surface water specimens from Lu Chao Harbor were subjected to incubation with lignin dissolved in solution as the exclusive carbon source at ambient temperature for a period of three weeks, aiming to augment the microbial community. Following this enrichment, the microbes were cultivated on marine 2216E-agar medium, leading to the selection of a distinct colony, designated as strain LCG002. This particular strain produced a colony that was white, circular, and featured regular, slightly elevated edges. Examination through transmission electron microscopy (TEM) disclosed that the cellular dimensions of strain LCG002 spanned 3.2–4.8 μm in length and 0.5–0.6 μm in width. Consistent with the characterization of non-flagellated species, such as *M. donghaensis*, *M. niveibacter*, and *M. aquimaris*, strain LCG002 was observed to lack flagella (Appendix A). The strain’s growth was found to be strictly aerobic, with optimal growth occurring in the marine 2216E medium at temperatures around 30 °C, and no growth was detected at temperatures below 10 °C (Appendix A). 

The major cellular fatty acids of strain LCG002 (>5.0%) included Feature 8 (C18:1ω7c and/or C18:1ω6c; 56.02%), C18:1 ω7c 11-methyl (20.71%), C16:0 (8.03%) and C18:0 (7.51%, Appendix A). The polar lipids of strain LCG002 were composed of phosphatidylglycerol (PG) and phosphatidylethanolamine (PE, Appendix A). The respiratory quinone of strain LCG002 was confirmed to be ubiquinone Q-10, with no detection of menaquinone in this strain.

### 3.2. The Phylogenetic Characteristics of Strain LCG002

Strain LCG002 contains three homogenous 16S rRNA gene sequences. The alignments of 16S rRNA gene sequences showed that strain LCG002 is closely related to *Marivivens aquimaris* GSB7, *Ketogulonicigenium* sp. NBU2586, *Ketogulonicigenium vulgare* WSH-001, *Aestuarium zhoushanense* G7 [27] (reclassified as *Marivivens donghaensis* G7) and *Marivivens* sp. JLT3646, with BLASTn identities of 96.38%, 96.13%, 96.03%, 96.03% and 96.03%, respectively. Furthermore, we acquired the complete genomic sequence of strain LCG002 and constructed a phylogenetic tree based on 120 conserved protein sequences (known as GTDB taxonomy). The analysis revealed that strain LCG002 formed a coherent phylogenetic cluster associated with *Marivivens aquimaris* GSB7, *Marivivens donghaensis* HF1, *Marivivens niveibacter* MCCC 1A06712, *Marivivens geojensis* FJ12, *Marivivens* sp. JLT3646, “*Aestuarium*” *zhoushanense* G7, *Marivivens donghaensis* CECT 8947 and *Marivivens donghaensis* KCTC 42776 (Figure 1).

Moreover, strain LCG002 shared ANIb values of 73.59–75.0 with the other eight *Marivivens* strains, surpassing the values observed with the nearest related genus, *Yoonia* (71.77–73.14) (Appendix A). The AAI values between strain LCG002 and the other *Marivivens* strains were 70.22 to 72.4 but were, at most, 68.65 with the *Yoonia* species (Appendix A). The DDH values between strain LCG002 and the other *Marivivens* strains were merely 14.6 to 15.5 (Appendix A). Consequently, strain LCG002 may be classified as a new species within the *Marivivens* genus, phylogenetically situated near the root of the *Marivivens* lineage.

### 3.3. Description of Genomic Features

The complete genome of strain LCG002 is composed of a single chromosome spanning a total of 2,934,017 base pairs (bp). The G + C content of strain LCG002 genome is 58%. The genomic content includes 3109 genes, which encompass 3021 protein-coding sequences, 52 transfer RNAs (tRNAs), three sets of ribosomal RNA (rRNA) operons, three non-coding RNAs (ncRNAs), and 25 pseudogenes (Table 1).

The genome’s graphical depiction is presented in Appendix A. Following the COG classification, a total of 2628 (86.99%) genes encoding proteins were assigned to 22 categories, as detailed in Table 2. The predominant COG categories included amino acid transport and metabolism (10.04%); general function prediction only (8.79%); translation, ribosomal structure and biogenesis (7.38%); carbohydrate transport and metabolism (7.34%); and energy production and conversion (6.81%). Moreover, up to seven genomic islands were identified within the genome of strain LCG002.

### 3.4. The Metabolic Characteristics of Strain LCG002

In our investigation of the metabolic traits and ecological capabilities of strain LCG002, we delineated the metabolic pathways and conducted a comparative analysis with other strains within the *Marivivens* genus (Figure 2). We found that LCG002 contains the pathways for utilization of many kinds of carbohydrates, carboxylic acids, amino acids, extracellular proteins, lignin and aromatic monomers as its carbon sources. In addition, it possesses the ability to utilize hydrogen and carbon monoxide as energy sources. Moreover, strain LCG002 exhibits versatile utilization profiles for nitrogen, phosphorus and sulfur sources. 

#### 3.4.1. The Utilization of Carbohydrates

The LCG002 contains 193 COG-Gs (carbohydrate transport and metabolism), including 34 glycoside hydrolases. However, none of these glycoside hydrolase genes are extracellular (signalP-fused) enzymes, suggesting that they may intake extracellular sugars through sugar transport proteins. Indeed, strain LCG002 possesses the ABC transport systems of alpha-1,4-digalacturonate (*augE*/*F*/*G*), trehalose/maltose (*thuE*/*F*/*G*/*K*), multiple sugar (*chvE* and *gguA*/*B*), glucose/mannose (*gstA*/*B*/*C* and *msmX*), ribose (*rbsA*/*B*/*C*/*D*), fructose (*frcA*/*B*/*C*), inositol (*ibpB* and *iatP*/*A*), alpha-glucoside (*agpA*/*B*/*C*/*D*), and sn-Glycerol 3-phosphate (ug*pA*/*B*/*C*/*E*). Moreover, all *Marivivens* strains contain the *smoE*/*F*/*G*/*K* genes, enabling them to transport sorbitol and mannitol. Specifically, compared to the other eight *Marivivens* strains, strain LCG002 possesses the transport systems of maltose/maltodextrin (*malE*/*F*/*G*/*K*), lactose/L-arabinose (*lacE*/*F*/*G*/*K*), as well as pectin-derived oligosaccharide transport system (*yesO*/*P*/*Q*, Appendix A). However, it lacks the alpha-glucoside transport system (*aglE*/*F*/*G*/*K*) and D-xylose transport system (*xylF*/*H*/*G*), which are present in other *Marivivens* strains.

Moreover, strain LCG002 demonstrates broad carbohydrate utilizing enzymes, efficiently metabolizing a range of monosaccharides, such as glucose, fructose, ribose, galactose and xylose, disaccharides, including chitobiose, lactose, melibiose, and sucrose, as well as oligosaccharides, like stachyose, raffinose, and manninotriose, and polysaccharides, like nystose. Our experiment further validated that strain LCG002 is capable of metabolizing a range of carbon sources, including fructose, sucrose, raffinose, nystose, and cellulose when provided individually (Figure 3). These pieces of evidence indicate its adaptability to diverse sugar substrates in marine ecosystems.

#### 3.4.2. Metabolism of Amino Acids and Extracellular Proteins

In terms of amino acid biosynthesis, all *Marivivens* strains are predicted to be capable of synthesizing all 20 types of amino acids. Although strain LCG002 loses the gene encoding acetylornithine deacetylase (EC:3.5.1.16) compared to other *Marivivens* strains, it is still able to convert N-acetylornithine and L-glutamate into L-ornithine through the action of glutamate N-acetyltransferase (EC:2.3.1.35). Therefore, strain LCG002 could be as capable of synthesizing arginine as other *Marivivens* strains. For amino acid metabolism, the degrading pathways of valine, leucine, isoleucine, phenylalanine, histidine and tryptophan are incomplete in almost all *Marivivens* strains, suggesting that amino acids are not the primary source of energy for this genus. 

In terms of the intake of amino acids and their derivatives, the ABC transport systems of general L-amino acids, polar amino acids, branched-chain amino acids, oligopeptides and dipeptides were predicted in strain LCG002. Moreover, both the genome and plasmid could encode the glycine betaine/proline transport system (Appendix A), allowing it to uptake glycine betaine from the environment, although it is unable to synthesize betaine due to a strain-specific loss of betaine–aldehyde dehydrogenase (EC:1.2.1.8, Appendix A). Additionally, six genes of extracellular (signalP-fused) peptidases were identified in strain LCG002, belonging to families M23, M48, S1 and S41. Our experiments further confirmed that strain LCG002 could grow with peptone as the sole carbon source (Figure 3). These pieces of evidence imply that proteolytic activity may constitute a viable metabolic strategy for *Marivivens* strains.

#### 3.4.3. Utilization of Carboxylates

All *Marivivens* strains display a diverse set of specific genes that are involved in the transport and metabolism of various carboxylates as carbon sources. This genetic profile includes genes for a cation/acetate symporter, sodium-dependent bicarbonate transport family permease, C4-dicarboxylate ABC transport system (*dctB*/*D*/*P*/*Q*/*M*), and tricarboxylic ABC transport systems (*tctA*/*B*/*C*/*D*/*E*, Appendix A). The widespread presence of acetyl–CoA synthetase (EC:6.2.1.1) and propionyl–CoA synthetase (EC:6.2.1.17) across *Marivivens* strains further highlights their ability to metabolize acetate and propionate.

Moreover, strain LCG002 contains strain-specific genes for 4-hydroxybutyrate dehydrogenase (EC:1.1.1.61, Appendix A), indicating its capability to utilize 4-hydroxybutyrate. Strain LCG002 also harbors strain-specific genes for formate dehydrogenase (EC:1.17.1.9, Appendix A), a tungstate-dependent enzyme that oxidizes formate to carbon dioxide. Accordingly, we identified strain-specific tungstate transporter genes (*tupA*/*B*/*C*) located approximately 4.5 kilobases upstream of the formate dehydrogenase genes (Appendix A).

Our experiment proved that strain LCG002 could utilize sodium acetate, sodium formate and sodium propionate as the sole carbon sources to sustain its growth (Figure 3). These findings collectively suggested that strain LCG002 has the potential to utilize a broad range of carboxylates as carbon sources.

#### 3.4.4. Lignin Degradation and Strain-Specific Utilization of Aromatic Acids

To ascertain the ligninolytic capacity of strain LCG002, we cultivated it under conditions where lignin was the exclusive carbon source. A significant increase in cell density was observed, which is indicative of the strain’s ability to degrade lignin (Appendix A). Lignin degradation is initiated through the synergistic action of various enzymes, such as lignin peroxidase (LiP), manganese peroxidase (MnP), laccase (Lac), and dye-decolorizing peroxidases (DyP), which are responsible for the decolorization of aniline blue [44,45,46]. Within the genome of strain LCG002, we identified a gene encoding a dyp-type peroxidase enzyme and observed robust decolorization upon incubation on 2216E-agar plates containing aniline blue (Figure 4). This observation substantiates that strain LCG002 is capable of producing enzymes that facilitate lignin degradation.

Furthermore, genome analysis showed that strain LCG002 has the potential to degrade aromatic monomers, including benzoate and 4-hydroxybenzoate, which is similar to strain JLT3646 as reported. In detail, strain LCG002 harbors benzoyl-CoA 2, 3-epoxidase (EC:1.14.13.208), 3-hydroxybenzoate 6-monooxygenase (EC:1.14.13.24), and benzoyl-CoA-dihydro-diol lyase (EC:4.1.2.44), which catalyze a series of reactions, including the ring-opening oxidation of benzoate, leading to conversion of benzoate to 3,4-didehydroadipyl-coA semialdehyde. Moreover, p-hydroxybenzoate 3-monooxygenase (EC:1.14.13.2) and protocatechuate 3,4-dioxygenase (EC:1.13.11.3) may facilitate strain LCG002 to transform 4-hydroxybenzoate into 3-carboxy-cis, cis-muconate. This intermediate is further metabolized through a series of enzymatic reactions involving 3-carboxy-cis, cis-muconate cyclo-isomerase (EC:5.5.1.2), 4-carboxymuconolactone decarboxylase (EC:4.1.1.44), and 3-oxoadipate enol-lactonase (EC:3.1.1.24), culminating in the entry of 3-oxoadipate into the tricarboxylic acid (TCA) cycle for complete oxidation. 

More interestingly, our genomic comparison identified two strain-specific gene clusters in strain LCG002, QQG91_RS01875 to QQG91_RS02065 and QQG91_RS14350 to QQG91_RS14370, which are predicted to encode 44 genes, including 20 strain-specific genes involved in the degradation of various aromatic acids (Appendix A). In detail, the first gene cluster harbors a strain-specific benzoate-CoA ligase (EC:6.2.1.25) involved in hydroxylation of 3-hydroxybenzoate to 2,5-dihydroxybenzoate, which is then processed by enzymes typically involved in tyrosine metabolism. Accordingly, the complete pathways of tyrosine metabolism are also strain LCG002-specific, whereas other *Marivivens* species lack the genes encoding homo-gentisate 1,2-dioxygenase (EC:1.13.11.5), maleylacetoacetate isomerase (EC:5.2.1.2), and fumarylacetoacetase (EC:3.7.1.2). Furthermore, the second gene cluster encodes strain-specific phenylacetate-CoA ligase (EC:6.2.1.30), ring-1,2-phenylacetyl-CoA epoxidase (EC:1.14.13.149), and oxepin-CoA hydrolase (EC:3.3.2.12), which suggests that LCG002 could be a potential phenylacetate degrader.

To further validate the aromatic acid degradation capabilities of LCG002, we conducted cultivation using benzoate, 3-hydroxybenzoate, and 4-hydroxybenzoate as the sole carbon sources in the flasks. Then, the bacterial cultures in these flasks were spread onto 2216E agar plates on day 0 (before cultivation) and day 7 (after cultivation). It is evident that, after 7 days of culture in the shake flask, the number of bacterial colonies on the plates significantly increased. These results provided compelling evidence for the degradation of aromatic acids by strain LCG002, which is hypothesized to be a result of horizontal gene transfer events (Figure 5).

#### 3.4.5. Utilization of Carbon Monoxide and Hydrogen

In addition to utilizing the aforementioned organic compounds as energy sources, strain LCG002 is also capable of oxidizing inorganic substances to generate energy, such as hydrogen and carbon monoxide. Aerobic carbon monoxide dehydrogenase (CODH) is a complex enzyme with multiple subunits that are involved in the oxidation of carbon monoxide (CO) to carbon dioxide (CO_2_) in some aerobic bacteria, e.g., *Oligotropha carboxidovorans*, *Hydrogenophaga pseudoflava* [47] and *Stappia stellulata* [48]. In strain LCG002, we have identified two operons encoding aerobic carbon monoxide dehydrogenase subunits (*coxG*/*L*/*M*/*S*). Furthermore, the second operon contains a *coxD* gene, which belongs to the APE2220 subfamily of the *MoxR* family. *CoxD* is crucial for the assembly of the (CuSMoO_2_) cluster in the carbon monoxide dehydrogenase and is vital for the bacterium’s survival and metabolic activity, especially in environments with high concentrations of CO [49,50].

For utilization of hydrogen, we found that all *Marivivens* species (but none of *Yoonia* species) have a large gene cluster (QQG91_RS12160 to QQG91_RS12255, Appendix A) encoding Ni/Fe-hydrogenase subunits (HypC/D/E) [51], post-translational modifier (*HyaD*) [52] and B-type cytochrome subunits (*HyaC* and *HybC*) [53]. Such a gene cluster also encodes *HupR*, *HupU*, *HupV*, and *HupT*, which regulate the uptake and integration of the hydrogenase enzyme into the cell membrane, as well as *HypB* and *HybF* [51], which facilitate nickel incorporation into the hydrogenase active site. The genes of hydrogenase and CODH can provide *Marivivens* species with an alternative energy source during periods of low oxygen or starvation, allowing these organisms to buffer against environmental changes and disturbances in marine ecosystems.

#### 3.4.6. The Acquirement of Nitrogen, Phosphorus and Sulfur Sources

Strain LCG002 exhibits a versatile nitrogen utilization profile, as indicated by the presence of genes encoding for various nitrogen transporters and assimilation pathways (Appendix A). The urea transporter (*urtA*/*B*/*C*/*D*/*E*) and ammonium transporter (amt) genes suggest the microorganism’s ability to uptake and utilize urea and ammonium, respectively, as nitrogen sources. The spermidine/putrescine transporter (*potA*/*B*/*C*/*D*) and the putrescine transporter (*potF*/*I*/*G*/*H*) genes indicate the microorganism’s capacity to import polyamines. Additionally, the general nucleoside transport system (*nupA*/*B*/*C*) implies the microorganism’s capability to take up nucleosides, which can serve as precursors for nucleic acid synthesis. Furthermore, the presence of the nitrate/nitrite transport system (*nrtA*/*B*/*C*) and genes involved in assimilatory nitrate reduction (*nasA*/*B*/*C*/*D*/*E*) reveals its ability to utilize nitrate and nitrite as alternative nitrogen sources. Overall, these genetic features enable the microorganism to thrive in various ecological niches where different nitrogen sources are available.

For phosphorus utilization, strain LCG002 harbors the transport systems of phosphate, sn-glycerol3-phosphate and phosphonate, while it also loses the gene encoding phospholipid ABC transport (*malCDEF*), which is conserved in the other eight *Marivivens* species.

For sulfur utilization, strain LCG002 demonstrates a robust metabolic profile, encompassing complete pathways for assimilatory sulfate reduction, as evidenced by the presence of the *cysCDJHIN* gene cluster. Furthermore, LCG002 is equipped with a full pathway for the utilization of dimethyl-sulfoniopropionate (DMSP), indicated by the *dmdABCD* genes. Moreover, different from *Yoonia* species that employ sulfite dehydrogenase (*soeABC*) to oxidize sulfite to sulfate, all *Marivivens* species harbor the SOX system (*soxABCDXYZ*), which typically transfers electrons to oxygen during the thiosulfate oxidation process [54]. This metabolic diversity highlights the adaptability of LCG002 to various sulfur sources, contributing to its ecological versatility and potential for survival in diverse environments.

## 4. Conclusions

In conclusion, strain LCG002, isolated from Lu Chao Harbor’s intertidal seawater, belongs to the genus *Marivivens*. It shows many strain-specific genes and a remarkable metabolic diversity of degrading lignin as well as lignin-derived aromatics, such as benzoate, 4-hydroxybenzoate, 3-hydroxybenzoate and phenylacetate. It is also capable of utilizing a wide array of carbon sources, such as carbohydrates, proteins and carboxylates, as well as inorganic gases, like hydrogen and carbon monoxide. Its nitrogen metabolism, supported by nitrate/nitrite and urea transporters, as well as its assimilatory nitrate reductase, ensures nitrogen availability. Additionally, the strain’s sulfur assimilation strategies, including the SOX system and the utilization of dimethyl-sulfoniopropionate (DMSP), demonstrate its versatility in sulfur acquisition. Collectively, LCG002’s metabolic capabilities and genetic makeup highlight its potential for biotechnological applications and its crucial role in the cycling of nutrients in marine environments, particularly in the degradation of lignocellulosic material and aromatic monomers.

## Figures and Tables

**Figure 1 microorganisms-12-01308-f001:**
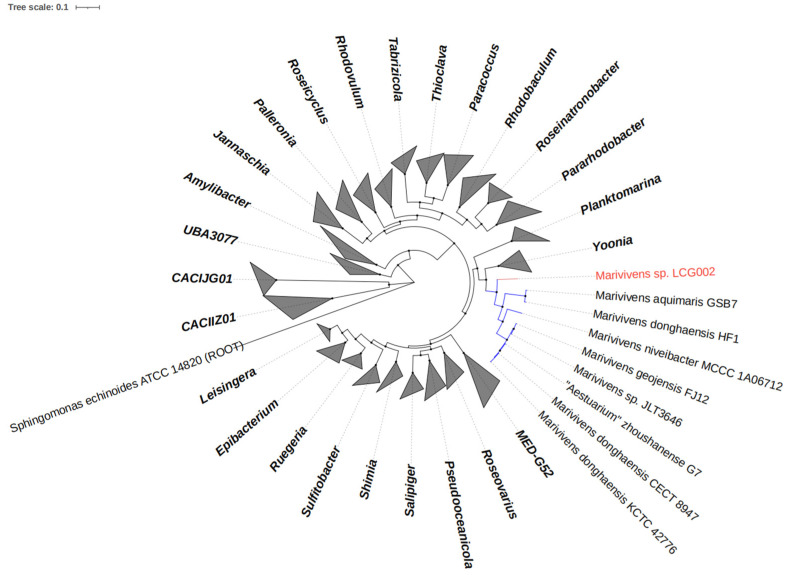
Phylogeny of the *Marivivens* species and their related species based on 120 concentrated proteins. The *Marivivens* sp. LCG002 and other *Marivivens* strains are highlighted, respectively, in red and blue.

**Figure 2 microorganisms-12-01308-f002:**
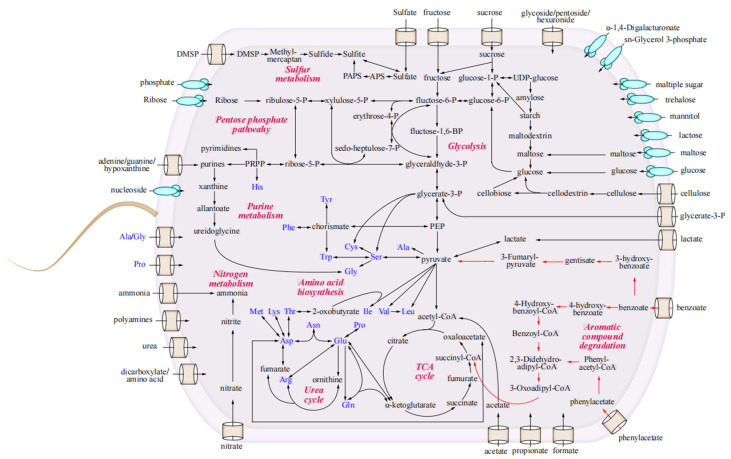
Metabolic potentials of strain LCG002. The black arrows denote metabolic pathways predicted in strain LCG002 and the red arrows highlight the pathways in strain LCG002 that are absent in other *Marivivens* strains.

**Figure 3 microorganisms-12-01308-f003:**
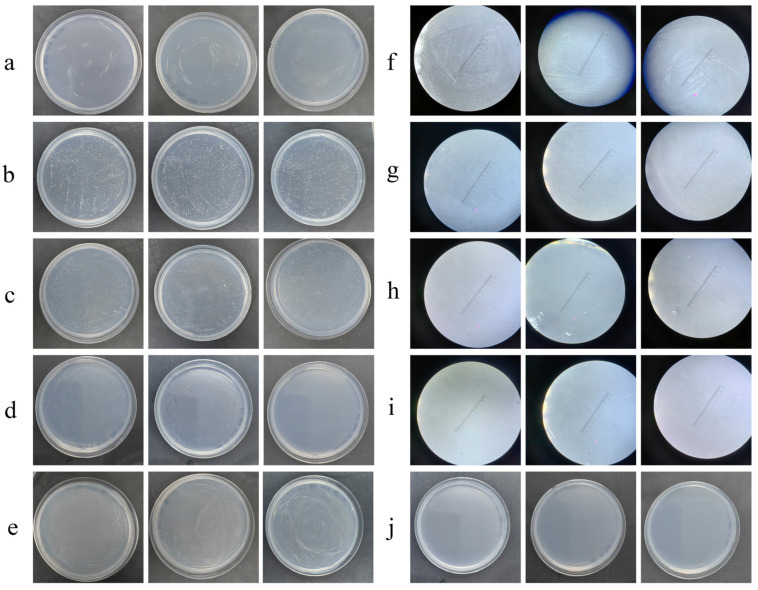
The carbon source utilization of strain LCG002. This represents ASW agar plates with (**a**) fructose, (**b**) raffinose, (**c**) nystose, (**d**) sodium acetate, (**e**) peptone, (**f**) sucrose, (**g**) cellulose, (**h**) sodium formate and (**i**) sodium propionate as the sole carbon source, respectively. (**j**) represents the blank control. Notably, (**f**–**i**) were captured using an 8× stereoscopic microscope because the colonies of LCG002 were relatively small when grown on agar plates containing these substrates. Each sample was tested with three independent biological replicates.

**Figure 4 microorganisms-12-01308-f004:**
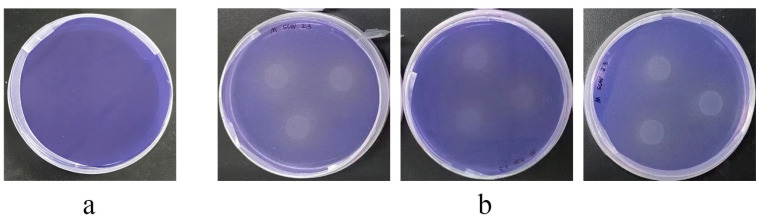
Decolorization of aniline blue by strain LCG002. (**a**) Blank control, and (**b**) decolorization of strain LCG002. Each sample was tested with three independent biological replicates.

**Figure 5 microorganisms-12-01308-f005:**
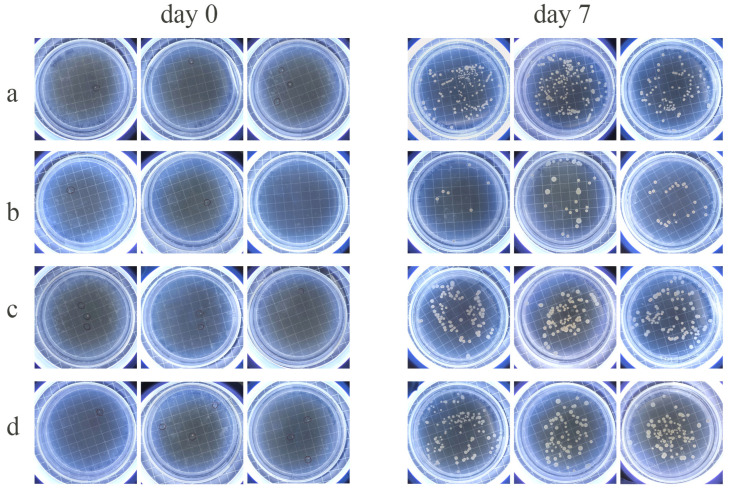
The aromatic acids and lignin utilization of strain LCG002. (**a**–**d**) represent 2216E agar plates coated with strain culture containing benzoate, 3-hydroxybenzoate, 4-hydroxybenzoate and lignin as the exclusive carbon source, respectively. The subfigures on the left column represent 2216E agar plates coated with culture incubated for 0 days, and those on the right represent that with culture incubated for 7 days. Each sample was tested with three independent biological replicates.

**Table 1 microorganisms-12-01308-t001:** Genome features of strain LCG002.

Items	Description
Size (bp)	2,934,017
G + C content (%)	58
Total Genes	3109
Protein-coding genes	3021
Genes assigned to COG	2628
rRNA operons	3
tRNA genes	52
ncRNA genes	3
Pseudogene	25
Gene islands	7

**Table 2 microorganisms-12-01308-t002:** The COG classification of strain LCG002.

COG Letter	COG Classification	Gene Number
J	Translation, ribosomal structure and biogenesis	194
K	Transcription	162
L	Replication, recombination and repair	89
D	Cell cycle control, cell division, chromosome partitioning	28
V	Defense mechanisms	41
T	Signal transduction mechanisms	93
M	Cell wall/membrane/envelope biogenesis	118
N	Cell motility	35
W	Extracellular structures	6
U	Intracellular trafficking, secretion, and vesicular transport	30
O	Posttranslational modification, protein turnover, chaperones	124
X	Mobilome: prophages, transposons	16
C	Energy production and conversion	179
G	Carbohydrate transport and metabolism	193
E	Amino acid transport and metabolism	264
F	Nucleotide transport and metabolism	83
H	Coenzyme transport and metabolism	154
I	Lipid transport and metabolism	136
P	Inorganic ion transport and metabolism	104
Q	Secondary metabolites biosynthesis, transport and catabolism	77
R	General function prediction only	231
S	Function unknown	161

## Data Availability

The GenBank accession number for the complete genome sequence of strain LCG002 is CP 127165.1.

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
