# Peer review of "The Phylogeny and Metabolic Potentials of an Aromatics-Degrading Marivivens Bacterium Isolated from Intertidal Seawater in East China Sea"

_microorganisms, 2024, doi:10.3390/microorganisms12071308_

Round 1

Reviewer 1 Report

Comments and Suggestions for Authors

Dear Authors,
Thank you for allowing me to read this interesting manuscript on the properties of lignin-degrading microorganisms isolated from seawater.
In my opinion, the overall quality of the prepared manuscript and its scientific content are far above the average, but before publication, I'd consider improving two minor imperfections of the result presentation.
Figure 3 description: It is not obligatory, but for better "readability" of this figure, I suggest replacing the a-j letters with the vertical texts with the sugar/sample names; I assume that there are three parallel replications shown as the three plates. 
Figure 5 description: as in the case of Figure 3 - I'd consider introducing a horizontal description above the two columns of the plate photos described as 0 days and 7 days (however, it is unclear why the 0-day incubated plates were shown?) Instead of a-h letters, I'd consider using the compound name or the abbreviations.  

I consider the above points only imperfections, but if the author agrees with them, they may be improved.

The manuscript's text, together with the presented data and the supplementary materials, composes an overall good manuscript, which should be accepted for publication.

Reviewer 2 Report

Comments and Suggestions for Authors

The article reports the characteristics of a novel strain of bacteria (Marivivens sp. LCG002), isolated from the seawater of Lu Chao Harbor in the East China Sea. Based on the study, the genomic comparison to other Marivivens species, the reconstruction of metabolic pathways, and the metabolic capabilities of LCG002 strain were presented. The discovered novel strain has a great potential to utilize lignin, thus serving the possibility to degrade lignocellulosic materials in marine ecosystems.

The manuscript is interesting and generally well-written. However, some issues should be resolved before it is accepted for publication. My comments are listed below:

1. Line 68 – the statement “we present the discovery of novel strains” is misleading. The Authors did detailed characteristics of the novel bacterium species and it was presented in the article, not discovery. The mentioned sentence should be corrected.

2. Line 229 – it is not clear what are the presented data 96.38, 96.13, 96.03, 96.03 and 96.03? The percentage of identity with other species? If so, the unit should be included.

3. Line 230 – please refer to the Figure number instead of “see below”

4. Chapter 3.4 – please discuss the results shown in Fig. 2.

5. Line 291- fructose is not disaccharide - please correct

6. Figure 3 – The results on the photos “d”,”g”,”h”,”I” are unclear. “d” looks similar to “j” (control plates). ”,”g”,”h”,”I” don’t look like agar plates. I can not see the bacteria cultures on them. Please improve.

7. Line 391 – “respectively”, should be added
